# Efficacy and safety of Atezolizumab plus Bevacizumab and Lenvatinib as first-line systemic therapies for hepatocellular carcinoma: A real-world study

Sangdao Boonkaya[1,2], Chidkamon Pattarawongpaiboon[1], Panarat Thaimai[1], Prooksa Ananchuensook[1,3,4], Supachaya Sriphoosanaphan[1,3,4], Suebpong Tanasanvimon[5], Nattaya Teeyapun[5], Nussara Pakvisal[5], Nipaporn Siripon[4], Sombat Treeprasertsuk[1], Piyawat Komolmit[4], Kessarin Thanapirom[1,3]*

1 Division of Gastroenterology, Department of Medicine, Faculty of Medicine, Chulalongkorn University and King Chulalongkorn Memorial Hospital, Thai Red Cross Society, Bangkok, Thailand, 2 Department of Internal Medicine, Queen Savang Vadhana Memorial Hospital, Chonburi, Thailand, 3 Center of Excellence in Hepatic Fibrosis and Cirrhosis, Chulalongkorn University, Bangkok, Thailand, 4 Excellence Center in Liver Diseases, King Chulalongkorn Memorial Hospital, Thai Red Cross Society, Bangkok, Thailand, 5 Division of Oncology, Department of Medicine, Faculty of Medicine, Chulalongkorn University, Bangkok, Thailand

* kessarin.t@chula.ac.th

## Abstract

Atezolizumab plus bevacizumab (ATEZO/BEV) and Lenvatinib (LEN) have demonstrated efficacy as first-line systemic therapies for unresectable hepatocellular carcinoma (HCC). This study aimed to compare the efficacy, safety, and outcomes of these two treatments. Data were retrospectively collected from 163 patients with unresectable HCC receiving first-line Lenvatinib (LEN) (n = 85) or ATEZO/BEV (n = 78) between 2020 and 2023 at Chulalongkorn University Hospital in Bangkok, Thailand. The primary outcome was overall survival (OS) following treatment. Propensity score matching (PSM) was used for analysis. The median patient age was 60.6 (SD 11.8) years; 82.2% were male. Most had hepatitis B/C (66.9%), BCLC stage C (73%), and Child-Pugh class A cirrhosis (63.8%). After PSM analysis, overall survival (OS) was significantly longer in the ATEZO/BEV group (12.7 vs. 7.5 months; p = 0.016, HR = 0.618, 95% CI: 0.417–0.916). However, there was no significant difference in progression-free survival (PFS) between ATEZO/BEV and LEN groups (10.8 vs. 7.8 months; p = 0.26, HR = 0.73, 95% CI: 0.431–1.255). Assessed by mRECIST, neither objective response rate (ORR: 23.7% vs. 19.7%, p = 0.555) nor disease control rate (DCR: 36.8% vs. 38.2%, p = 0.867) differed significantly. Multivariate analysis revealed alpha-fetoprotein ≥500 ng/mL (HR = 1.881, 95% CI: 1.028–3.443, p = 0.04), tumor size (HR = 1.833, 95% CI: 1.010–3.327, p = 0.046) and ATEZO/BEV therapy (HR = 0.604, 95%CI: 0.373–0.977, p = 0.04) were independently associated with OS. Adverse event rates were comparable (ATEZO/BEV 43.7% vs. LEN 56.3%; p = 0.08).

**Data availability statement:** All raw data files are available from the figshare database (URL: https://figshare.com/s/0dcff780565bd8302ca9).

**Funding:** This work was supported by Ratchadapiseksompotch Endowment Fund of the Center of Excellence in Hepatic Fibrosis and Cirrhosis Research Unit (GCE 3300170037 to KT).

**Competing interests:** The authors have declared that no competing interests exist.

In conclusion, the study demonstrates that ATEZO/BEV significantly improves OS compared to LEN in patients with unresectable HCC, despite similar PFS, ORR, and DCR. Both treatments have comparable safety profiles.

## Introduction

Hepatocellular carcinoma (HCC) is one of the most prevalent cancers worldwide and the third leading cause of cancer-related mortality [1]. The incidence of HCC has been steadily rising, particularly in regions with high hepatitis B and C infection rates, as well as in areas with an increasing prevalence of metabolic dysfunction-associated steatotic liver disease (MASLD) and cirrhosis [1]. Over the past decade, the management of HCC has significantly improved due to a deeper understanding of its natural history, advances in staging techniques, refined treatment algorithms, and the development of novel therapeutic options. However, HCC remains one of the most challenging cancers to treat.

Systemic therapies remain the cornerstone of treatment for patients with unresectable HCC, providing essential options to extend survival and improve quality of life [2]. In recent years, the landscape of systemic treatment for HCC has undergone a significant transformation, with multiple novel therapies targeting various aspects of cancer biology. These include multikinase inhibitors, monoclonal antibodies, and immune checkpoint inhibitors (ICIs), each with distinct mechanisms of action and clinical benefits. Multikinase inhibitors, such as sorafenib and lenvatinib (LEN), target multiple signaling pathways involved in tumor growth and angiogenesis, providing effective treatment options for advanced HCC [2,3]. Monoclonal antibodies, including bevacizumab (BEV), directly target angiogenesis by inhibiting vascular endothelial growth factor (VEGF). This disrupts the formation of blood vessels that supply oxygen and nutrients to tumors [4]. Recently, ICIs have garnered significant attention for their potential in treating HCC. These therapies—including atezolizumab (ATEZO), pembrolizumab, and the combination of tremelimumab plus durvalumab—work by blocking immune checkpoints that tumors exploit to evade immune detection. This reactivates the immune system, enabling it to recognize and attack cancer cells. ICIs have demonstrated promising results in various cancers, and their approval for HCC has ushered in a new era in the treatment of this challenging disease [5]. These novel therapies have not only improved survival outcomes but also revolutionized the management of unresectable HCC, offering hope to patients who previously had limited therapeutic options.

The combination of ATEZO and BEV has been approved as first-line therapy for unresectable HCC following the results of the IMbrave150 trial [6]. The study demonstrated that ATEZO/BEV significantly improved both progression-free survival (PFS) and overall survival (OS) compared to sorafenib. Prior to the approval of ATEZO/BEV, LEN was approved as a first-line therapy for unresectable HCC based on the REFLECT trial, which demonstrated non-inferior survival outcomes compared to sorafenib [3]. The potential efficacy of LEN compared to other systemic treatments

remains an area of interest. There is limited evidence on the efficacy and safety of LEN compared to ATEZO/BEV. Therefore, this study aims to evaluate and compare the therapeutic efficacy and safety profiles of ATEZO/BEV versus LEN as first-line systemic treatments for patients with unresectable HCC.

## Patients and methods

### Patients

This retrospective study consecutively included 163 patients with unresectable HCC who received first-line therapy with either LEN (n = 85) or ATEZO/BEV (n = 78) between January 2020 and December 2023 at King Chulalongkorn Memorial Hospital, Bangkok, Thailand. Eligible participants met the following criteria: 1) Adults aged 18 years or older with a confirmed diagnosis of unresectable HCC, based on typical imaging or histology [7], who received either LEN or ATEZO/BEV as first-line systemic therapy; 2) Child-Pugh class A or B liver function; and 3) An Eastern Cooperative Oncology Group (ECOG) performance status of 0 or 1. We excluded patients with: 1) a second primary cancer; 2) an ECOG performance status of 3–4; and 3) a Child-Pugh score > 9. Inclusion and exclusion criteria were systematically applied to all eligible patients identified through electronic medical records during the study period.

This study did not obtain informed consent from participants because it was conducted retrospectively using data extracted from the hospital's electronic medical records. This study was conducted in accordance with the ethical principles outlined in the Declaration of Helsinki. This study was approved by the Ethics Committee and Institutional Review Board (IRB) of Chulalongkorn University Hospital (IRB No. 0311/67).

Data were extracted from the hospital's electronic medical records. To protect patient confidentiality, the final research dataset was fully de-identified, and no personal identifiers were included in the analysis or reporting.

### Design and treatment

Propensity score matching (PSM) was used to balance covariate distributions between treatment groups. In this study, propensity scores were calculated using a logistic regression model including the following baseline factors as covariates: age, sex, body weight (BW), etiology of chronic liver disease, Child-Pugh class, albumin-bilirubin (ALBI) grade, AFP level, and BCLC stage. A one-to-one nearest-neighbor matching approach was employed, using an optimal caliper of 0.2 and without replacement, to match patients between two groups, each comprising 76 individuals.

In the ATEZO/BEV group, patients received a combination ATEZO (1,200 mg) and BEV (15 mg/kg) administered intravenously every three weeks. In the LEN group, patients received oral LEN (12 mg/day for those with a BW ≥ 60 kg or 8 mg/day for those with a BW < 60 kg). If any unacceptable adverse events (AEs) occurred—as defined by the National Cancer Institute's Common Terminology Criteria for Adverse Events—dose reductions or treatment interruptions were implemented until symptoms resolved to below Grade 2. The AEs were retrospectively extracted from electronic health records.

### Treatment response assessment

Therapeutic responses were assessed through dynamic contrast-enhanced CT or MRI every 12–16 weeks until death or treatment discontinuation. Treatment responses were assessed according to the modified Response Evaluation Criteria in Solid Tumors (mRECIST) [8]. The radiological evaluations were performed by two independent radiologists blinded to clinical outcomes, with a third radiologist adjudicating any discrepancies. Objective response rates (ORRs) were defined as the combined percentage of patients achieving a complete response (CR) or partial response (PR), while disease control rates (DCRs) were defined as the percentage of patients with CR, PR, or stable disease (SD). Overall survival (OS) was defined as the time from the date of treatment initiation to the date of death from any cause, whereas progression-free survival (PFS) was defined as the time from treatment initiation to the date of documented disease progression or death, whichever occurred first. The data cut-off for this analysis was June 30, 2024. For the OS analysis, data for patients who

were alive at the data cut-off date were censored at that date. Patients lost to follow-up before this date were censored at their last known date of contact. The proportion of patients alive in each treatment arm at the time of the final data cut-off was calculated to assess data maturity.

### Statistical analysis

Categorical variables were expressed as numbers and percentages and compared between groups using Pearson's chi-square test or Fisher's exact test. Continuous variables were expressed as mean±standard deviation and compared between groups using the Mann–Whitney $U$ test.

To minimize potential selection bias, one-to-one PSM was used. PFS and OS were estimated using the Kaplan–Meier method, with differences assessed by the log-rank test. Multivariate analyses were conducted using the Cox proportional hazards model to identify variables associated with OS. Statistical analyses were performed using IBM® SPSS Statistics, Version 22 (Armonk, NY, USA), with two-sided p-values<0.05 considered statistically significant.

## Results

### Baseline patient characteristics before PSM

The baseline characteristics of the 163 patients are presented in Table 1. Of these, 134 (82.2%) were male, with a mean age of 60.6±11.8 years (range: 30–86 years). The mean baseline BW was 64.6±11.8 kg. The most common comorbidities included type 2 diabetes mellitus in 44 patients (27%), ischemic heart disease in 9 patients (5.5%), and chronic kidney disease in 3 patients (1.8%). Hepatitis B virus infection was the most common cause of chronic liver disease, accounting for 74 cases (45.4%). Cirrhosis was classified as Child-Pugh class A in 104 patients (63.8%) and class B in 60 patients (36.2%). Albumin-bilirubin (ALBI) scores of grade 2 and grade 3 were observed in 103 (63.2%) and 19 (11.7%) patients, respectively. According to the BCLC staging system, 44 patients (27%) were classified as stage B and 119 patients (73%) as stage C. Infiltrative lesions were present in 27 patients (16.6%), macrovascular invasion (MVI) was detected in 64 patients (39.3%), and extrahepatic metastasis was observed in 76 patients (46.9%). Regarding prior therapies, transarterial chemoembolization (TACE) was the most common, having been performed in 91 patients (55.8%). Other previous treatments included microwave or radiofrequency ablation (MWA/RFA) (11%, n=18), surgical resection (11%, n=18), external radiation therapy (XRT) (14.7%, n=24), and Yttrium-90 (Y-90) radioembolization (6.7%, n=11). LEN was administered to 85 patients (52.1%), while 78 (47.9%) received ATEZO/BEV. There were no significant differences in baseline characteristics between the ATEZO/BEV and LEN groups (Table 1). S1 Fig presents the CONSORT flowchart of patient enrollment.

### Efficacy prior to PSM

The median follow-up was 9.98 months (8.9 months in the LEN group and 11.1 months in the ATEZO/BEV group). Table 2 presents the patients' clinical responses during therapy. No significant differences were observed in ORR (17.6% vs. 23.1%, p=0.39) or DCR (36.5% vs. 35.9%, p=0.06) between the ATEZO/BEV and LEN groups. Moreover, median PFS did not differ significantly between the ATEZO/BEV and LEN groups (10.8 months vs. 7.8 months; p=0.36, HR=1.27, 95% CI: 0.76–2.15) (Fig 1A). In contrast, median OS was significantly longer in the ATEZO/BEV group than in the LEN group (12.25 months vs. 7.49 months; p=0.036, HR=0.66, 95%CI: 0.45–0.97) (Fig 1B).

### Efficacy after PSM

A total of 152 patients remained after matching. The PSM analysis yielded 76 matched pairs. There were no significant differences in baseline characteristics between the two groups, as shown in S1 Table. No significant differences were observed in ORR (19.7% vs. 23.7%; p=0.55) or DCR (38.2% vs. 36.8%; p=0.86) between the ATEZO/BEV and LEN

**Table 1. Baseline characteristics.**

| Clinical characteristic | LEN N = 85 (%) | ATEZO/BEV N = 78 (%) | P-value |
|---|---|---|---|
| Mean age, years | 61.5 ± 10.6 | 59.8± 13.0 | 0.41 |
| Male | 70 (82.3%) | 64 (82.1%) | 0.05 |
| ECOG performance status<br>ECOG 0<br>ECOG 1<br>ECOG 2 | <br>35 (41.2%)<br>41 (48.2%)<br>9 (10.6%) | <br>43 (55.1%)<br>34 (43.6%)<br>1 (1.3%) | 0.05 |
| BW, kg ± SD | 63.84 ± 12.12 | 65.44 ± 12.74 | 0.27 |
| T2DM | 20 (23.5%) | 24 (30.8%) | 0.30 |
| CKD | 3 (3.5%) | 0 | 0.09 |
| IHD | 4 (4.7%) | 5 (6.4%) | 0.63 |
| Etiology<br>Hepatitis B virus<br>Hepatitis C virus<br>Non-viral | <br>38 (44.7%)<br>21 (24.7%)<br>26 (30.6%) | <br>36 (46.2%)<br>13 (16.7%)<br>28 (35.9%) | 0.47 |
| ALBI<br>Grade 1<br>Grade 2<br>Grade 3 | <br>22 (25.9%)<br>50 (58.8%)<br>13 (15.3%) | <br>19 (24.4%)<br>53 (67.9%)<br>6 (7.7%) | 0.27 |
| Child-Pugh Score<br>Score <8<br>Score ≥8 | <br>71 (83.5%)<br>14 (16.5%) | <br>71 (91%)<br>7 (9%) | 0.12 |
| Albumin, g/dL | 3.49 ± 0.61 | 3.53 ± 0.51 | 0.43 |
| Total bilirubin, mg/dL | 1.42 ± 1.13 | 1.41 ± 1 | 0.72 |
| AFP, ng/mL<br>    <500<br>    ≥ 500 | <br>61 (71.8)<br>24 (28.2) | <br>44 (58.7)<br>31 (41.3) | 0.08 |
| Sodium, mEq/L | 135.63 ± 4.15 | 135.45 ± 4.46 | 0.96 |
| BCLC<br>Stage B<br>Stage C | <br>27 (31.8%)<br>58 (68.2%) | <br>17 (21.8%)<br>61 (78.2%) | 0.15 |
| Previous treatment | 59 (69.4%) | 58 (74.4%) | 0.48 |
| MWA/RFA | 14 (16.5%) | 4 (5.1%) | 0.02* |
| Resection | 7 (8.2%) | 11 (14.1%) | 0.23 |
| TACE | 52 (61.2%) | 39 (50%) | 0.15 |
| XRT | 10 (11.8%) | 14 (17.9%) | 0.26 |
| Y-90 | 2 (2.4%) | 9 (11.5) | 0.02* |
| Maximum tumor diameter, cm | 7.44 ± 5.59 | 8.16 ±5.96 | 0.48 |
| Macrovascular invasion | 32 (37.6%) | 32 (41) | 0.48 |
| Infiltrative lesion | 14 (16.5%) | 13 (16.7%) | 0.97 |
| Extrahepatic metastasis | 35 (41.2%) | 41 (53.2%) | 0.12 |
| Median follow-up time, months | 5.72 (0.92-33.54) | 8.69 (0.69-36.93) | 0.05 |

Categorical variables were expressed as number and frequency, while continuous variables were expressed as mean±SD or median (interquartile range)

Abbreviation: ATEZO/BEV; Atezolizumab plus bevacizumab, AFP; alpha-fetoprotein, ALBI; albumin-bilirubin score, BCLC; Barcelona Clinic liver Cancer, BW; body weight, CKD; chronic kidney disease, ECOG; Eastern Cooperative Oncology Group, IHD; ischemic heart disease, LEN; Lenvatinib, MWA/RFA; microwave ablation or radiofrequency ablation, PD; progressive disease, T2DM; type 2 diabetes mellitus, TACE; transarterial chemoembolization, XRT; radiation therapy, Y-90; Yttrium-90 radioembolization

**Table 2. Clinical response of patients during therapy.**

|  | LEN N = 85 (%) | ATEZO/BEV N = 78 (%) | p-value |
|---|---|---|---|
| **Complete response** | 1 (1.3%) | 0 | 0.014 |
| **Partial response** | 14 (18.4%) | 18 (23.7%) | 0.014 |
| **Stable disease** | 17 (22.4%) | 16 (21.1%) | 0.014 |
| **Progressive disease** | 44 (57.9%) | 42 (55.3%) | 0.014 |
| **Objective response rate** | 15 (17.6%) | 18 (23.1%) | 0.389 |
| **Disease control rate** | 31 (36.5%) | 28 (35.9%) | 0.939 |

Abbreviation: ATEZO/BEV; Atezolizumab plus bevacizumab, LEN; Lenvatinib

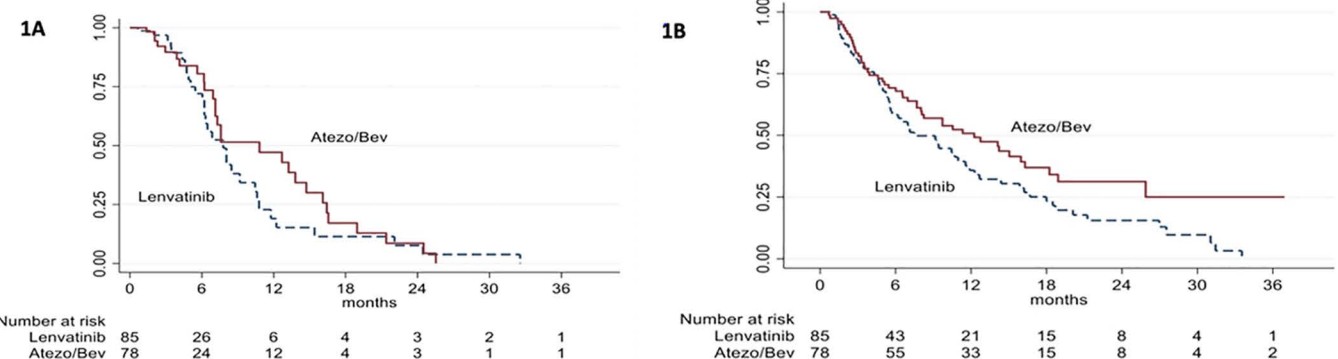

**Fig 1. Comparison of the ATEZO/BEV and LEN groups before PSM. (A)** Progression-free survival; **(B)** Overall survival. Abbreviations: ATEZO/BEV, atezolizumab plus bevacizumab; LEN, Lenvatinib; PSM, propensity-score matching.

groups. S2 Table shows the clinical response of patients during therapy after PSM. The median PFS did not differ significantly between the ATEZO/BEV and LEN groups (10.8 vs. 7.8 months; p = 0.26, HR = 0.73, 95% CI: 0.431–1.255) (Fig 2A). The median OS was significantly longer in the ATEZO/BEV group than in the LEN group (12.7 vs. 7.5 months; p = 0.016, HR = 0.618, 95% CI: 0.417–0.916) (Fig 2B).

The standardized mean differences (SMD) for baseline covariates before and after matching were shown in S3 Table. The SMD for all baseline covariates was substantially reduced after matching. The absolute SMD for nearly all covariates was well below the recommended threshold of 0.1, indicating that a good balance was achieved between both groups and that systematic imbalances were adequately eliminated.

Analysis of post-progression management revealed significant differences between the treatment arms. Patients who progressed on ATEZO/BEV were significantly more likely to receive subsequent systemic therapy compared to those on LEN (35.7% vs. 18.2%; P = 0.04). Conversely, a significantly higher proportion of patients in the Lenvatinib group received only best supportive care (52.2% vs. 30.9%; P = 0.01). The use of locoregional salvage interventions was comparable between ATEZO/BEV and LEN groups (33.3% vs. 29.5%, p = 0.57).

## Adverse events after PSM

AEs of any grade occurred in 67 patients (56.3%) receiving LEN and 52 patients (43.7%) receiving ATEZO/BEV, with no significant difference observed after PSM (p = 0.081). The prevalence of anorexia (21.1% vs. 6.6%, p = 0.004), diarrhea (13.2% vs. 0%, p = 0.001), and fatigue (32.9% vs. 10.5%, p = 0.001) was higher in the LEN group than in the ATEZO/BEV group. Table 3 presents the AEs in both groups. The rate of treatment-related AEs causing discontinuation was similar in

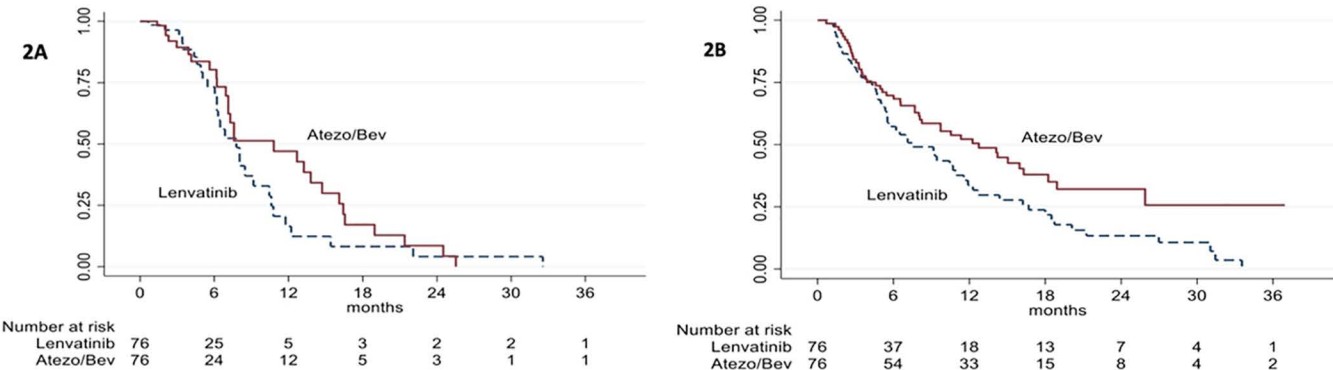

**Fig 2. Comparison of the ATEZO/BEV and LEN groups after PSM. (A)** Progression-free survival; **(B)** Overall survival. Abbreviations: ATEZO/BEV, atezolizumab plus bevacizumab; LEN, Lenvatinib; PSM, propensity-score matching.

**Table 3. Treatment-related adverse events.**

|  | Lenvatinib N = 76 (%) | ATEZO/BEV N = 76 (%) | P-value |
|---|---|---|---|
| **Anorexia** | 16 (21.1) | 5 (6.6) | 0.010* |
| **Nausea/Vomiting** | 3 (3.9) | 2 (2.6) | 0.649 |
| **Diarrhea** | 10 (13.2) | 0 | 0.001* |
| **Fatigue** | 25 (32.9) | 8 (10.5) | 0.001* |
| **Hand-foot syndrome reaction** | 7 (9.2) | 3 (3.9) | 0.191 |
| **Adrenal insufficiency** | 2 (2.6) | 0 | 0.155 |
| **Hypothyroid** | 3 (3.9) | 6 (7.9) | 0.303 |
| **Proteinuria** | 6 (7.9) | 10 (13.2) | 0.290 |
| **Hypertension** | 10 (13.2) | 6 (7.9) | 0.290 |
| **Abdominal pain** | 1 (1.3) | 3 (3.9) | 0.311 |
| **GI bleeding** | 3 (3.9) | 8 (10.5) | 0.118 |
| **Liver injury** | 2 (2.6) | 5 (6.6) | 0.246 |
| **Colitis** | 0 | 2 (2.6) | 0.204 |
| **Pneumonitis** | 0 | 1 (0.1) | 0.342 |
| **Hepatitis** | 20 (26.3) | 13 (17.1) | 0.168 |
| **Cytopenia** | 0 | 2 (2.6) | 0.155 |
| **Treatment-related adverse events leading to discontinuation** | 6 (7.9) | 7 (9.2) | 0.628 |
| **Treatment-related adverse events leading to dose reduction** | 6 (7.9) | 0 | 0.027 |

both groups. While treatment-related AEs led to dose reduction in 7.9% of patients on Lenvatinib, no patients on ATEZO/BEV required a dose reduction (p = 0.027).

## Factors associated with overall survival after PSM

Univariate analysis identified several factors associated with OS, including BW changes before and after treatment, Child-Pugh scores ≥ 8, ALBI grades 2 or 3, AFP levels ≥ 500 ng/mL, presence of infiltrative lesions, tumor size, and albumin levels. Multivariate analysis demonstrated that treatment with ATEZO/BEV, an AFP level ≥500 ng/dL, and tumor size were independently associated with OS (Table 4).

**Table 4. Factor associated with OS in patients with unresectable HCC treated with atezolizumab/bevacizumab or lenvatinib after propensity-score matching.**

| | Univariate analysis | | | Multivariate analysis | | |
|---|---|---|---|---|---|---|
| | HR | 95%CI | P-value | HR | 95%CI | P-value |
| **Treatment (LEN, ATEZO/BEV)** | 0.618 | 0.417-0.916 | 0.016* | 0.604 | 0.373-0.977 | 0.040* |
| **Age (<70, ≥ 70 years)** | 0.856 | 0.518-1.415 | 0.544 | | | |
| **Gender (Male, Female)** | 1.002 | 0.607-1.652 | 0.995 | | | |
| **Difference BW** | 1.029 | 1.001-1.047 | 0.001* | 1.022 | 1-1.045 | 0.051 |
| **T2DM** | 0.882 | 0.564-1.378 | 0.581 | | | |
| **CKD** | 0.449 | 0.062-3.234 | 0.427 | | | |
| **IHD** | 0.844 | 0.370-1.929 | 0.688 | | | |
| **Etiology (Viral vs non-viral)** | 0.893 | 0.586-1.359 | 0.597 | | | |
| **Albumin** | 0.409 | 0.276-0.608 | <0.001* | 0.824 | 0.463-1.466 | 0.51 |
| **Platelet** | 0.915 | 0.624-1.127 | 0.652 | | | |
| **NLR** | 1.012 | 0.541-1.254 | 0.612 | | | |
| **ALBI**<br>  Grade 1 vs 2<br>  Grade 2 vs 3 | <br>0.517<br>2.339 | <br>0.313-0.856<br>1.323-4.138 | <br>0.010*<br>0.003* | <br>1.52<br>1.275 | <br>0.803-2.876<br>0.328-4.953 | <br>0.198<br>0.726 |
| **Child-Pugh Score (<8, ≥8)** | 3.543 | 2.070-6.065 | <0.001* | 2.557 | 0.684-9.558 | 0.163 |
| **AFP (<500, ≥500 ng/dL)** | 1.797 | 1.195-2.701 | <0.001* | 1.881 | 1.028-3.443 | 0.040* |
| **Previous local HCC treatment** | 1.463 | 0.948-2.257 | 0.086 | | | |
| **BCLC (stage B vs stage C)** | 0.844 | 0.544-1.310 | 0.449 | | | |
| **Tumor diameter (<10, ≥ 10 cm)** | 1.798 | 1.205-2.685 | 0.004* | 1.833 | 1.01-3.327 | 0.046* |
| **Macrovascular invasion** | 1.297 | 0.861-1.953 | 0.213 | | | |
| **Infiltrative lesion** | 1.806 | 1.099-2.967 | 0.020* | 1.623 | 0.843-3.123 | 0.147 |
| **Extrahepatic metastasis** | 0.771 | 0.517-1.150 | 0.202 | | | |

Abbreviation: AFP; alpha-fetoprotein, ALBI; albumin-bilirubin score, ATEZO/BEV; atezolizumab/bevacizumab, BCLC; Barcelona Clinic liver Cancer, BW; body weight, CKD; chronic kidney disease, HCC; hepatocellular carcinoma, IHD; ischemic heart disease, LEN; Lenvatinib, NLR; neutrophil-to-lymphocyte ratio, T2DM; type 2 diabetes mellitus

## Discussion

In this study, we compared the efficacy and safety of ATEZO/BEV versus LEN as first-line therapies for unresectable HCC. The main findings were that ATEZO/BEV showed superior efficacy, yielding significantly longer OS than LEV in both unadjusted and PSM-adjusted analyses. After PSM adjustment for confounding factors, the median OS was 12.7 months with ATEZO/BEV versus 7.5 months with LEN. However, there were no significant differences in PFS, ORR, or DCR between the two groups in either analysis. Moreover, the incidence of gastrointestinal AEs was more common with LEN treatment.

Regarding treatment efficacy, the IMbrave150 trial demonstrated that ATEZO/BEV improved PFS (median 12.6 vs 8.6 months) and OS (median 25.8 vs 21.9 months) compared to sorafenib in patients with unresectable HCC [6]. In contrast, the REFLECT trial demonstrated that LEN achieved a median OS (13.6 vs 12.3 months) that was non-inferior to sorafenib while also providing longer PFS (7.4 vs 3.7 months). (3) To date, no prospective, randomized controlled trials have compared the efficacy of ATEZO/BEV and LEN in patients with unresectable HCC. However, three prior PSM studies have assessed their efficacy, and the results were inconsistent. Niizeki et al. reported significantly longer median OS (not reached vs. 20.2 months, p = 0.039) and PFS (8.3 vs. 6.0 months, p = 0.005) in the ATEZO/BEV group compared to the LEN group, despite no significant difference in ORR between the two groups (44.8% vs. 46.7%, p = 0.64) [9]. In contrast, Kim et al. found no significant differences in median OS (not reached vs. 12.8 months; p = 0.357) or PFS (5.7 vs.

6.0 months; p = 0.738) between the ATEZO/BEV and LEN groups. Similarly, there was no significant difference in ORR between the ATEZO/BEV and LEN groups (32.6% vs. 31.5%; p = 0.868). (10) Maesaka *et al.* reported significantly longer median PFS in the ATEZO/BEV group than in the LEN group (8.8 vs. 5.2 months, p = 0.012). However, there were no significant differences between the two groups in terms of median OS (not reached vs. 20.6 months; p = 0.577) or ORR (43.8% vs. 52.4%; p = 0.330). (11) The current study demonstrated that median OS was significantly longer in the ATEZO/ BEV group than in the LEN group (12.7 vs. 7.5 months; p = 0.016). There was no significant difference in median PFS (10.8 vs. 7.8 months; p = 0.26) or ORR (23.7% vs. 19.7%; p = 0.555) between the ATEZO/BEV and LEN groups. Variations in patient characteristics and tumor-related factors may account for the discrepancies among these studies. In the study by Kim BK *et al.*, 86.2% of patients were classified as BCLC stage C, 51.3% had MVI, and 75% had viral hepatitis as the underlying cause. In the study by Niizeki *et al.*, 50.7% of patients had BCLC stage C disease, 17.4% had MVI, and 52.2% had viral hepatitis as the underlying cause. In the study by Mesaka K. *et al.*, 46.2% of patients were classified as BCLC stage C, 17.4% had MVI, and 55.3% had viral hepatitis as the underlying etiology. This study included 73% of patients with BCLC stage C disease, 65.8% of patients with HBV-related liver disease and 40.1% with MVI. In contrast, Niizeki *et al.* reported that only 15.5% of patients had HBV-related liver disease, with 44% classified as BCLC stage C and the remaining 56% in stages A or B. Additionally, only 10% had macrovascular invasion. The shorter OS and PFS observed in our study compared to IMBRAVE 150 and REFLECT study might be attributable to the more advanced baseline liver disease and tumor burden. Furthermore, the relatively short follow-up period might result in immature survival data, as evidenced by the Kaplan-Meier curves for OS, which decline steeply without plateauing. This highlights the need for longer-term follow-up to confirm the durability of the observed survival benefit.

*The prolonged OS observed in the ATEZO/BEV group compared to the LEN group in this study may be attributed to several factors. The combination of an immune checkpoint inhibitor (ICI) with an anti-angiogenic agent may enhance anti-tumor immune responses while simultaneously inhibiting tumor vascularization. This dual mechanism may lead to superior long-term survival outcomes [10]. Another possible explanation involves socioeconomic factors. In real-world settings, particularly in developing countries, access to systemic therapies remains limited. In Thailand, for example, reimbursement policies do not cover systemic therapies, forcing most patients to pay out of pocket. ATEZO/BEV costs approximately 3,854 USD every three weeks, whereas LEN costs around 1,182 USD per month. In our clinical practice, the selection of systemic treatment is often influenced by the patient's financial situation. A significant disparity was observed in the receipt of second-line systemic therapy: 35.7% of patients who progressed on ATEZO/BEV received subsequent treatment, compared to only 18.2% of those in the LEN group (P = 0.04). This difference likely reflects financial barriers that limited access to more expensive second-line options for patients who had failed first-line LEN. Therefore, we believe that socio-economic status may influence treatment decisions and could contribute to the observed disparities in survival outcomes [11,12].

Regarding AEs, there were no significant differences in adverse events and overall rates of treatment discontinuation between LEN and ATEZO/BEV groups; their AE profiles varied significantly. Patients receiving LEN treatment had a significantly higher incidence of decreased appetite (21.1% vs. 6.6%; P = 0.010), diarrhea (13.2% vs. 0%; P = 0.001), and fatigue (32.9% vs. 10.5%; P = 0.001). Furthermore, treatment-related AEs resulted in significantly more dose reductions for the LEN group compared to the ATEZO/BEV groups (7.9% vs 0%, p = 0.027). Additionally, hand-foot syndrome, adrenal insufficiency, and hypothyroidism occurred more frequently in LEN-treated patients, though these differences were not statistically significant. Moreover, there was a trend toward a higher incidence of proteinuria and hepatic injury in the ATEZO/ BEV group, although these differences were not statistically significant.

This study has several limitations that should be acknowledged. First, the study had a retrospective design, which introduced inherent biases despite the application of PSM. Second, the generalizability of our findings is limited by the single-center design. The study cohort is from Thailand and has a high prevalence of viral hepatitis as the underlying etiology for HCC (79.4%). Given that this differs significantly from Western populations, where non-viral HCC is more

common, caution is advised when extrapolating our results to patients with different underlying liver diseases. Third, while our PSM analysis adjusted for several key clinical variables, we acknowledge the potential for residual confounding from unmeasured factors. Baseline liver stiffness measurement was not included in the model due to inconsistent data availability. Moreover, the reliability of liver stiffness measured by transient elastography can be compromised in this patient population, as the values may be confounded by the tumor, especially in patients with a high tumor burden or infiltrative lesions. Fourth, A formal analysis of the duration of response (DoR) was precluded by the limited number of patients who achieved a complete or partial response. Finally, our study did not incorporate an analysis of biological or immune-based biomarkers to stratify patient outcomes. The future of HCC management undoubtedly lies in biomarker-driven treatment algorithms that can predict which patients are most likely to benefit from specific systemic therapies [13]. The retrospective design of our study precluded the systematic collection of tissue for assessing the tumor immune microenvironment, and key circulating biomarkers like des-gamma-carboxy prothrombin were not consistently available. Future prospective, real-world studies are urgently needed to integrate these multi-faceted biomarker analyses with clinical data to create more precise, individualized treatment strategies.

In summary, our study demonstrates that ATEZO/BEV provides a significant survival advantage over LEN in patients with unresectable HCC, improving OS despite comparable ORR and DCR. ATEZO/BEV and LEN demonstrated similar overall safety profiles. Based on this finding, LEN should be an alternative option in patients with contraindication or unaffordability to ATEZO/BEV.

## Supporting information

**S1 Fig. Consort diagram of patients' enrollment.**
(TIF)

**S1 Table. Baseline characteristics after propensity score matching.**
(DOCX)

**S2 Table. The clinical response of patients during therapy after PSM.**
(DOCX)

**S3 Table. Standardized mean differences for baseline covariates before and after matching.**
(DOCX)

## Acknowledgments

We thank the staff of the Division of Gastroenterology and Hepatology, Excellence Center in Liver Diseases, Center of Excellence in Hepatic Fibrosis and Cirrhosis, Faculty of Medicine, Chulalongkorn University, and King Chulalongkorn Memorial Hospital, Thai Red Cross Society, for their technical assistance and clinical support.

## Author contributions

**Conceptualization:** Piyawat Komolmit, Kessarin Thanapirom.

**Data curation:** Sangdao Boonkaya, Chidkamon Pattarawongpaiboon, Kessarin Thanapirom.

**Formal analysis:** Sangdao Boonkaya.

**Investigation:** Sangdao Boonkaya, Panarat Thaimai, Prooksa Ananchuensook, Supachaya Sriphoosanaphan, Suebpong Tanasanvimon, Nattaya Teeyapun, Nussara Pakvisal, Nipaporn Siripon, Sombat Treeprasertsuk, Piyawat Komolmit.

**Project administration:** Sangdao Boonkaya.

**Resources:** Piyawat Komolmit, Kessarin Thanapirom.

**Supervision:** Suebpong Tanasanvimon, Sombat Treeprasertsuk, Piyawat Komolmit, Kessarin Thanapirom.

**Validation:** Kessarin Thanapirom.

**Writing – original draft:** Sangdao Boonkaya.

**Writing – review & editing:** Sangdao Boonkaya, Kessarin Thanapirom.

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
