## [Decision Letter · Decision Letter 0]

4 Sep 2025

PONE-D-25-42455Efficacy and safety of Atezolizumab plus Bevacizumab and Lenvatinib as first-line systemic therapies for hepatocellular carcinoma: a real-world studyPLOS ONE

Dear Dr. Thanapirom,

Thank you for submitting your manuscript to PLOS ONE. After careful consideration, we feel that it has merit but does not fully meet PLOS ONE’s publication criteria as it currently stands. Therefore, we invite you to submit a revised version of the manuscript that addresses the points raised during the review process.

**ACADEMIC EDITOR:**

Main methodological issues to be corrected:

- include more variables (baseline laboratory markers or liver stiffness measurements, concurrent or previous treatments, comorbidities...) in the model

- provide more information about the patients' follow-up  over 1 year after observation onset and about post-progression therapies (to improve reliability of prognosis data)

- detail response evaluation and interpretation criteria more deeply (to exclude interpretation biases due to the retrospective design of the study) 

Minor issues:

- change figures as requested

- English language editing

We look forward to receiving your revised manuscript.

Kind regards,

Carmelo Caldarella, Ph.D., M.D.

Academic Editor

PLOS ONE

Journal Requirements:

4. In the online submission form, you indicated that data cannot be shared publicly because they contain confidential patient information. Data are available from the corresponding author (contact via kessarin.t@chula.ac.th) for researchers who meet the criteria for access to confidential data and have appropriate ethical approval.

6. Please note that funding information should not appear in any section or other areas of your manuscript. We will only publish funding information present in the Funding Statement section of the online submission form. Please remove any funding-related text from the manuscript.

Additional Editor Comments :

Dear Authors, the Reviewers have carefully analyzed your manuscript and concluded that there are some methodological issues that require re-evaluation in order to make this paper more scientifically sound and clear to the reader, as well as to improve reliability and reproducibility of the results achieved.

Particularly, more variable (like baseline laboratory markers or liver stiffness measurements, concurrent or previous treatments, comorbidities) should be included in the model, and more information about the patients who survived longer than the first 12 months from observation onset and about post-progression therapies should be included, so that differences in prognosis are more reliable and clinically useful.

Moreover, response evaluation and interpretation criteria should be reported with more details, considering that retrospective studies are more prone to interpretation biases than prospective ones.

Some figures and English text editing is also desirable.

Waiting for the revised version of your manuscript

Best regards,

Carmelo Caldarella, PhD MD

Reviewers' comments:

Reviewer's Responses to Questions

**Comments to the Author**

1. Is the manuscript technically sound, and do the data support the conclusions?

Reviewer #1: Yes

Reviewer #2: Yes

2. Has the statistical analysis been performed appropriately and rigorously?

Reviewer #1: Yes

Reviewer #2: Yes

3. Have the authors made all data underlying the findings in their manuscript fully available?

Reviewer #1: Yes

Reviewer #2: Yes

4. Is the manuscript presented in an intelligible fashion and written in standard English?

Reviewer #1: Yes

Reviewer #2: Yes

5. Review Comments to the Author

Reviewer #1: This study demonstrates that ATEZO/BEV significantly improves OS compared to LEN in patients with unresectable HCC, despite similar PFS, ORR, and DCR. Both treatments have comparable safety profiles. I have some advice for you.

1.Figures have disappeared at the bottom of this manuscripts.

2.Some grammatical errors in articles should be revised.

Reviewer #2: The authors present a retrospective real-world analysis comparing atezolizumab plus bevacizumab to lenvatinib as first-line systemic therapies in patients with unresectable hepatocellular carcinoma. The clinical relevance of this work is undeniable, as head-to-head randomized controlled trials directly addressing this therapeutic comparison are lacking, and clinicians often rely on retrospective evidence and propensity score-matched analyses to guide treatment selection. The manuscript is clearly written, and the study benefits from a relatively well-characterized cohort and careful application of propensity score matching. The findings, particularly the survival advantage with atezolizumab plus bevacizumab despite similar progression-free survival and objective response rates, are potentially meaningful. Nevertheless, there are several important methodological issues and interpretative limitations that must be addressed before the work can be considered for publication.

The first major concern relates to the retrospective design and the inherent biases that arise despite the use of propensity score matching. The inclusion criteria, exclusion parameters, and data collection windows are not fully detailed. It is unclear whether all patients treated in the specified time frame were consecutively included, or whether certain subgroups might have been underrepresented. Without clear clarification, the risk of selection bias remains high. Moreover, although the authors applied one-to-one nearest-neighbor matching, the choice of a caliper of 0.2 and the covariates included in the logistic regression model may not be sufficient to account for important clinical confounders. For example, baseline laboratory markers such as platelet count, neutrophil-to-lymphocyte ratio, or liver stiffness measurements, which are increasingly recognized as important prognostic determinants in hepatocellular carcinoma, were not considered. Similarly, the presence of comorbidities, previous local therapies, or concomitant supportive care measures could have significantly impacted survival outcomes. A sensitivity analysis including additional variables, or at minimum a discussion of why these parameters were excluded, would strengthen the validity of the comparisons.

Another methodological limitation is the relatively short follow-up period, with a median under twelve months. The observed median overall survival in both arms is substantially lower than in pivotal clinical trials, which may reflect the advanced stage and higher tumor burden of the enrolled population, but may also stem from limited observation time and immature survival data. This discrepancy raises concerns about the robustness of the survival estimates. The authors should provide more granular details on censoring, the proportion of patients alive at last follow-up, and whether survival curves plateaued or continued to decline steeply beyond one year. Without this information, the survival advantage of atezolizumab plus bevacizumab over lenvatinib might be overestimated.

Treatment exposure and post-progression therapies also represent an area requiring greater detail. The discussion acknowledges that socioeconomic barriers influenced second-line treatment access, yet the manuscript does not provide clear data on how many patients in each arm received subsequent systemic therapy, locoregional salvage interventions, or best supportive care only. Given the well-documented impact of treatment sequencing on overall survival in hepatocellular carcinoma, the omission of these details limits interpretability. A breakdown of post-progression therapies, stratified by treatment arm, would allow readers to better understand whether the survival difference observed is attributable to first-line efficacy or subsequent management differences driven by cost and accessibility.

The methods for assessing response also warrant scrutiny. The manuscript reports that mRECIST criteria were applied, but it is not specified whether evaluations were conducted by blinded radiologists, whether inter-observer variability was assessed, or whether central review was performed. These details are not trivial, since retrospective real-world imaging assessments can be inconsistent, and progression-free survival may be particularly sensitive to variations in interpretation. Additionally, while objective response rate and disease control rate were reported, duration of response—an endpoint of growing importance in immunotherapy-based regimens—was not addressed. Including this analysis would have enhanced the understanding of how responses differ qualitatively between the two regimens.

The handling of alpha-fetoprotein and other tumor biomarkers deserves more thorough exploration. The multivariate analysis confirms alpha-fetoprotein ≥500 ng/mL and tumor size as independent predictors of poor overall survival, which is consistent with prior literature. However, des-gamma-carboxy prothrombin (DCP), another validated biomarker with both diagnostic and prognostic implications, was not included in the dataset. Given its widespread use in Asian populations, the omission is striking and should be at least acknowledged. Furthermore, alpha-fetoprotein dynamics during treatment, rather than baseline levels alone, could provide valuable insight into biological activity and treatment efficacy. A sensitivity analysis considering AFP response at eight to twelve weeks might have clarified whether biochemical response aligned with radiologic and survival outcomes.

The study presents safety data, but the adverse event reporting is somewhat superficial. It is not indicated whether adverse events were prospectively recorded in electronic health records or retrospectively extracted, and the grading system, while referenced, is not explicitly described in the results. Importantly, the lack of reporting on immune-related adverse events such as hepatitis, colitis, or pneumonitis, even if absent, is problematic when discussing immunotherapy-based combinations. Moreover, adverse events leading to dose reduction or permanent discontinuation should be highlighted, since tolerability and treatment persistence critically affect outcomes in real-world practice.

From a statistical standpoint, the use of propensity score matching is appropriate, but the presentation of results would benefit from additional clarity. Balance diagnostics, such as standardized mean differences for baseline covariates before and after matching, are not reported. Without this, it is difficult to assess whether the matching adequately eliminated systematic imbalances. Furthermore, the authors rely heavily on p-values for interpreting differences, yet confidence intervals are often wide and overlap substantially, particularly for progression-free survival. The discussion should place greater emphasis on the uncertainty of these estimates rather than drawing strong conclusions based on borderline statistical significance.

A broader issue is the limited generalizability of the findings. The cohort is derived from a single tertiary hospital in Thailand, with a very high prevalence of viral hepatitis as the underlying etiology. While this is representative of regional epidemiology, hepatocellular carcinoma in Western populations is increasingly driven by metabolic dysfunction-associated steatotic liver disease, which may respond differently to immunotherapy or anti-angiogenic agents. The authors should more clearly acknowledge that extrapolation of their findings to non-viral HCC should be cautious.

Finally, the study leaves unaddressed the pressing need for integrated biomarker-driven algorithms in hepatocellular carcinoma. Real-world comparative studies such as this one are valuable, but they risk remaining descriptive unless coupled with efforts to stratify patients more precisely based on molecular or immune signatures. The immunomicroenvironment of hepatocellular carcinoma is highly heterogeneous, with some tumors characterized by immune exclusion and others by immune infiltration. Atezolizumab plus bevacizumab might preferentially benefit the latter group through synergistic enhancement of anti-tumor immunity and vascular normalization. Incorporating assessments of immune infiltrates, PD-L1 expression, or circulating immune markers would have allowed the authors to begin addressing which patients derive the greatest benefit. Such analyses could ultimately feed into decision-support algorithms that combine clinical features, biomarkers such as AFP and DCP, and immunomicroenvironmental characteristics to guide individualized first-line therapy selection (please refer to PMID: 36901717 and expand).

In conclusion, this study makes a timely contribution to the literature by comparing atezolizumab plus bevacizumab with lenvatinib in unresectable hepatocellular carcinoma, and its findings align with the growing perception of immunotherapy-based combinations as superior in terms of survival benefit. However, methodological shortcomings, limited biomarker integration, and the absence of granular data on follow-up and subsequent therapies restrict the strength of the conclusions. Addressing these issues would substantially enhance the manuscript’s impact. Looking forward, the integration of tumor biology, circulating biomarkers, and immunomicroenvironment profiling into real-world datasets will be critical to developing robust algorithms that can guide screening, surveillance, and first-line treatment strategies in hepatocellular carcinoma.

6. PLOS authors have the option to publish the peer review history of their article (what does this mean?). If published, this will include your full peer review and any attached files.

Reviewer #1: No

Reviewer #2: No

---

## [Author Response · Author response to Decision Letter 1]

21 Oct 2025

Point-by-point response to reviewer comments

We sincerely appreciate all the valuable comments and suggestions which helped us improve the quality of the article. Below are the detailed, point-by-point responses to the reviewer’s comments. The manuscript has been updated for clarification and improvement

Additional Editor Comments :

Dear Authors, the Reviewers have carefully analyzed your manuscript and concluded that there are some methodological issues that require re-evaluation in order to make this paper more scientifically sound and clear to the reader, as well as to improve reliability and reproducibility of the results achieved.

Particularly, more variable (like baseline laboratory markers or liver stiffness measurements, concurrent or previous treatments, comorbidities) should be included in the model, and more information about the patients who survived longer than the first 12 months from observation onset and about post-progression therapies should be included, so that differences in prognosis are more reliable and clinically useful.

Moreover, response evaluation and interpretation criteria should be reported with more details, considering that retrospective studies are more prone to interpretation biases than prospective ones.

Some figures and English text editing is also desirable.

Waiting for the revised version of your manuscript

We thank the reviewers and editor for their helpful comments and have revised the manuscript to address the points raised. The entire manuscript has undergone a thorough professional language edit to improve clarity, grammar, and style. Furthermore, all figures have been reviewed and enhanced to ensure high resolution and clear labelling for better readability.

Reviewer comments:

Reviewer #1: This study demonstrates that ATEZO/BEV significantly improves OS compared to LEN in patients with unresectable HCC, despite similar PFS, ORR, and DCR. Both treatments have comparable safety profiles. I have some advice for you.

1.Figures have disappeared at the bottom of this manuscripts.

2.Some grammatical errors in articles should be revised.

We thank the reviewer for the comments. We sincerely apologize for this mistake. This was a technical error that occurred during the final file compilation. We have carefully checked the revised manuscript and can confirm that all figures are now correctly placed and clearly legible at the end of the document. The entire manuscript has been thoroughly proofread and edited to correct all grammatical errors and improve sentence structure.

Reviewer #2: The authors present a retrospective real-world analysis comparing atezolizumab plus bevacizumab to lenvatinib as first-line systemic therapies in patients with unresectable hepatocellular carcinoma. The clinical relevance of this work is undeniable, as head-to-head randomized controlled trials directly addressing this therapeutic comparison are lacking, and clinicians often rely on retrospective evidence and propensity score-matched analyses to guide treatment selection. The manuscript is clearly written, and the study benefits from a relatively well-characterized cohort and careful application of propensity score matching. The findings, particularly the survival advantage with atezolizumab plus bevacizumab despite similar progression-free survival and objective response rates, are potentially meaningful. Nevertheless, there are several important methodological issues and interpretative limitations that must be addressed before the work can be considered for publication.

We thank the reviewer for the effort in evaluating our manuscript. We are grateful for the reviewer's positive assessment of our study's clinical relevance, the clarity of the manuscript, and the careful application of our propensity score matching methodology. It is encouraging that the reviewer recognizes the potential significance of our findings, especially given the current lack of head-to-head randomized controlled trials.

We also agree with the reviewer that there are important methodological issues and interpretative limitations that require careful consideration. Guided by the reviewer’s insightful feedback, we have carefully revised the manuscript to address each of these concerns.

1. The first major concern relates to the retrospective design and the inherent biases that arise despite the use of propensity score matching. The inclusion criteria, exclusion parameters, and data collection windows are not fully detailed. It is unclear whether all patients treated in the specified time frame were consecutively included, or whether certain subgroups might have been underrepresented. Without clear clarification, the risk of selection bias remains high.

Response

Thank you for your suggestions. We agree that the retrospective design inherently introduces potential biases, even with the use of propensity score matching. We have acknowledged this limitation in the discussion section of the manuscript. In addition, to provide greater clarity, we confirm that our study was designed to minimize this risk of selection bias by including all consecutive patients with unresectable hepatocellular carcinoma who initiated first-line systemic therapy with either Atezolizumab plus Bevacizumab or Lenvatinib at our institution between 2020 and 2023. The inclusion and exclusion criteria were applied systematically to the electronic medical records of every patient who started these treatments during the study period. This consecutive enrollment approach ensures that our study population is representative of the real-world patient cohort, thereby reducing the likelihood of underrepresenting specific subgroups. We have added a sentence to the Methods section to clarify this point.

We have revised the discussion section.

“First, the study had a retrospective design, which introduced inherent biases despite the application of PSM.” (Page 14, line 5-7)

We have revised the Method Section.

“This retrospective study consecutively included 163 patients with unresectable HCC who received first-line therapy with either LEN (n = 85) or ATEZO/BEV (n = 78) between 1 January 2020 and 31 December 2023 at King Chulalongkorn Memorial Hospital, Bangkok, Thailand. Eligible participants met the following criteria: 1) Adults aged 18 years or older with a confirmed diagnosis of unresectable HCC, based on typical imaging or histology (7), who received either LEN or ATEZO/BEV as first-line systemic therapy; 2) Child-Pugh class A or B liver function; and 3) An Eastern Cooperative Oncology Group (ECOG) performance status of 0 or 1. We excluded patients with: 1) a second primary cancer; 2) an ECOG performance status of 3–4; and 3) a CTP score > 9. Inclusion and exclusion criteria were systematically applied to all eligible patients identified through electronic medical records during the study period.” (Page 4, line 7-17)

2. Moreover, although the authors applied one-to-one nearest-neighbor matching, the choice of a caliper of 0.2 and the covariates included in the logistic regression model may not be sufficient to account for important clinical confounders. For example, baseline laboratory markers such as platelet count, neutrophil-to-lymphocyte ratio, or liver stiffness measurements, which are increasingly recognized as important prognostic determinants in hepatocellular carcinoma, were not considered. Similarly, the presence of comorbidities, previous local therapies, or concomitant supportive care measures could have significantly impacted survival outcomes. A sensitivity analysis including additional variables, or at minimum a discussion of why these parameters were excluded, would strengthen the validity of the comparisons.

Response

We would like to thank you for your helpful suggestions. We agree that including additional covariates strengthens the model and helps to account for potential clinical confounders. In response to this feedback. We have added comorbidities (diabetes mellitus, ischemic heart disease and chronic kidney disease), a history of previous locoregional therapies, platelet count and NLR in the logistic regression analysis (Revised Table 4).

We acknowledge that liver stiffness is a valuable prognostic indicator. However, due to the retrospective nature of our study, these data were not uniformly documented for the entire cohort, which precluded their inclusion in the PSM analysis without introducing significant amounts of missing data. Furthermore, the liver stiffness values as evaluated by transient elastography can be confounded by the tumor burden, especially in patients with large or infiltrative HCC. We have mentioned these points in the limitations of the study.

We have revised Table 4.

We have revised the results section.

“The most common comorbidities included type 2 diabetes mellitus in 44 patients (27%), ischemic heart disease in 9 patients (5.5%), and chronic kidney disease in 3 patients (1.8%).” (Page 6, line 5-7)

“Regarding prior therapies, transarterial chemoembolization (TACE) was the most common, having been performed in 91 patients (55.8%). Other previous treatments included microwave or radiofrequency ablation (MWA/RFA) (11%, n=18), surgical resection (11%, n=18), external radiation therapy (XRT) (14.7%, n=24), and Yttrium-90 (Y-90) radioembolization (6.7%, n=11).” (Page 6, line 14-18)

We have revised the discussion section.

“Third, while our PSM analysis adjusted for several key clinical variables, we acknowledge the potential for residual confounding from unmeasured factors. Baseline liver stiffness measurement was not included in the model due to inconsistent data availability. Moreover, the reliability of liver stiffness measured by transient elastography can be compromised in this patient population, as the values may be confounded by the tumor, especially in patients with a high tumor burden or infiltrative lesions.” (Page 14, line 11-16)

3. Another methodological limitation is the relatively short follow-up period, with a median under twelve months. The observed median overall survival in both arms is substantially lower than in pivotal clinical trials, which may reflect the advanced stage and higher tumor burden of the enrolled population, but may also stem from limited observation time and immature survival data. This discrepancy raises concerns about the robustness of the survival estimates. The authors should provide more granular details on censoring, the proportion of patients alive at last follow-up, and whether survival curves plateaued or continued to decline steeply beyond one year. Without this information, the survival advantage of atezolizumab plus bevacizumab over lenvatinib might be overestimated.

Response

We thank the reviewer for raising this important point regarding the follow-up period and the observed survival outcomes. We agree that the relatively short follow-up period is a key limitation of our study and warrants a more detailed discussion. The discrepancy in median overall survival compared to the pivotal IMbrave 150 and REFLECT trials may be explained by a higher tumor burden in this study or limited observation time and immature survival data. We also acknowledge in our limitations that longer follow-up is necessary to obtain mature survival data.

We have revised the methodology section for more clarification. Overall survival was defined as the time from the date of treatment initiation to the date of death from any cause, whereas progression-free survival was defined as the time from treatment initiation to the date of documented disease progression or death, whichever occurred first. The data cut-off for this analysis was June 30, 2024. For the overall survival analysis, data for patients who were alive at the data cut-off date were censored at that date. Patients lost to follow-up before this date were censored at their last known date of contact. The proportion of patients alive in each treatment arm at the time of the final data cut-off was calculated to assess data maturity.

We have revised the method section.

“Overall survival (OS) was defined as the time from the date of treatment initiation to the date of death from any cause, whereas progression-free survival (PFS) was defined as the time from treatment initiation to the date of documented disease progression or death, whichever occurred first. The data cut-off for this analysis was June 30, 2024. For the OS analysis, data for patients who were alive at the data cut-off date were censored at that date. Patients lost to follow-up before this date were censored at their last known date of contact. The proportion of patients alive in each treatment arm at the time of the final data cut-off was calculated to assess data maturity.” (Page 5, line 16-23)

We have revised the discussion section.

“The shorter OS and PFS observed in our study compared to IMBRAVE 150 and REFLECT study, might be attributable to the more advanced baseline liver disease and tumor burden. Furthermore, the relatively short follow-up period might resulted in immature survival data, as evidenced by the Kaplan-Meier curves for OS, which decline steeply without plateauing. This highlights the need for longer-term follow-up to confirm the durability of the observed survival benefit.” (Page5, line 6-11)

4. Treatment exposure and post-progression therapies also represent an area requiring greater detail. The discussion acknowledges that socioeconomic barriers influenced second-line treatment access, yet the manuscript does not provide clear data on how many patients in each arm received subsequent systemic therapy, locoregional salvage interventions, or best supportive care only. Given the well-documented impact of treatment sequencing on overall survival in hepatocellular carcinoma, the omission of these details limits interpretability. A breakdown of post-progression therapies, stratified by treatment arm, would allow readers to better understand whether the survival difference observed is attributable to first-line efficacy or subsequent management differences driven by cost and accessibility.

Response

We thank the reviewer for this important point. We agree that post-progression therapies have a significant impact on overall survival and that a lack of these data can limit the interpretability of our findings. We have reviewed our electronic patient records to collect this information and have included it in the results section.

We have revised the method section

“Analysis of post-progression management revealed significant differences between the treatment arms. Patients who progressed on ATEZO/BEV were significantly more likely to receive subsequent systemic therapy compared to those on LEN (35.7% vs. 18.2%; P = 0.04). Conversely, a significantly higher proportion of patients in the Lenvatinib group received only best supportive care (52.2% vs. 30.9%; P = 0.01). The use of locoregional salvage interventions was comparable between ATE/BEV and LEN groups (33.3% vs. 29.5%, p=0.57).” (Page 9, line 10-15)

We have revised the discussion section

“A significant disparity was observed in the receipt of second-line systemic therapy: 35.7% of patients who progressed on ATE/BEV received subsequent treatment, compared to only 18.2% of those in the LEN group (P = 0.04). This difference likely reflects financial barriers that limited access to more expensive second-line options for patients who had failed first-line LEN.” (Page 13, line 22-26)

5.The methods for assessing response also warrant scrutiny. The manuscript reports that mRECIST criteria were applied, but it is not specified whether evaluations were conducted by blinded radiologists, whether inter-observer variability was assessed, or whether central review was performed. These details are not trivial, since retrospective real-world imaging assessments can be inconsistent, and progression-free survival may be particularly sensitive to variations in interpretation. Additionally, while objective response rate and disease control rate were reported, duration of response—an endpoint of growing importance in immunotherapy-based regimens—was not addressed. Including this analysis would have enhanced the understanding of how responses differ qualitatively between the two regimens.

Response

Than

---

## [Decision Letter · Decision Letter 1]

6 Nov 2025

Efficacy and safety of Atezolizumab plus Bevacizumab and Lenvatinib as first-line systemic therapies for hepatocellular carcinoma: a real-world study

PONE-D-25-42455R1

Dear Dr. Thanapirom,

We’re pleased to inform you that your manuscript has been judged scientifically suitable for publication and will be formally accepted for publication once it meets all outstanding technical requirements.

Kind regards,

Carmelo Caldarella, Ph.D., M.D.

Academic Editor

PLOS ONE

Additional Editor Comments (optional):

Dear Authors, I am pleased to inform you that the revised manuscript has been appreciated by our Reviewers and now it is suitable for publication in this journal.

Best regards,

Carmelo Caldarella

Reviewers' comments:

Reviewer's Responses to Questions

**Comments to the Author**

1. If the authors have adequately addressed your comments raised in a previous round of review and you feel that this manuscript is now acceptable for publication, you may indicate that here to bypass the “Comments to the Author” section, enter your conflict of interest statement in the “Confidential to Editor” section, and submit your "Accept" recommendation.

Reviewer #1: All comments have been addressed

Reviewer #2: All comments have been addressed

2. Is the manuscript technically sound, and do the data support the conclusions?

Reviewer #1: Yes

Reviewer #2: Yes

3. Has the statistical analysis been performed appropriately and rigorously?

Reviewer #1: Yes

Reviewer #2: Yes

4. Have the authors made all data underlying the findings in their manuscript fully available?

Reviewer #1: Yes

Reviewer #2: Yes

5. Is the manuscript presented in an intelligible fashion and written in standard English?

Reviewer #1: Yes

Reviewer #2: Yes

6. Review Comments to the Author

Reviewer #1: The study demonstrates that ATEZO/BEV significantly improves OS compared to LEN in patients with unresectable HCC, despite similar PFS, ORR, and DCR. Both treatments have comparable safety profiles.

Reviewer #2: The authors have clarified several of the questions I raised in my previous review. Most of the major problems have been addressed by this revision.

7. PLOS authors have the option to publish the peer review history of their article (what does this mean?). If published, this will include your full peer review and any attached files.

Reviewer #1: No

Reviewer #2: No

---

## [Editor Report · Acceptance letter]

PONE-D-25-42455R1

PLOS One

Dear Dr. Thanapirom,

I'm pleased to inform you that your manuscript has been deemed suitable for publication in PLOS One. Congratulations! Your manuscript is now being handed over to our production team.

Kind regards,

on behalf of

Dr. Carmelo Caldarella

Academic Editor

PLOS One